# Impact of an Interdisciplinary Educational Programme on Students’ Physical Activity and Fitness

**DOI:** 10.3390/healthcare11091256

**Published:** 2023-04-27

**Authors:** José Francisco Jiménez-Parra, Alfonso Valero-Valenzuela

**Affiliations:** SAFE (Salud, Actividad Física y Educación) Research Group, Department of Physical Activity and Sport, Faculty of Sport Sciences, University of Murcia, 30720 Murcia, Spain

**Keywords:** active breaks, physically active lessons, active learning, teaching personal and social responsibility, physical education, physical condition, health habits

## Abstract

Educational initiatives and actions are needed to provide students with skills to create active habits and lifestyles in order to increase the physical activity and fitness levels of young people. The main objective of this study was to analyse the effects of a classroom-based physical activity and life skills programme on students’ physical activity and fitness levels. The sample consisted of 65 students in the 6th grade of Primary Education, aged between 11 and 13 years (11.86 ± 0.53), divided into a control group (CG) and an experimental group (EG) for convenience and accessibility. This was a quasi-experimental, repeated measures, longitudinal cut-off research design. For 16 weeks, an interdisciplinary educational programme based on a combination of classroom-based physical activity and life skills was implemented. The results showed that the educational programme had a positive effect on students’ physical activity levels during school hours, specifically on reducing sedentary time (*p* < 0.001) and increasing light (*p* < 0.001) and moderate-to-vigorous (*p* < 0.001) physical activity. Positive effects were also found on the variables of explosive strength (*p* < 0.05) and aerobic capacity (*p* < 0.05) related to physical fitness. In conclusion, interdisciplinary educational programmes based on the promotion of physical activity and life skills could contribute to the development of responsible daily physical activity habits in students to facilitate their transfer to other environments (social and family) outside school.

## 1. Introduction

Nowadays, there is high concern among global bodies about meeting the third Sustainable Development Goal (SDG) proposed by the United Nations for 2030, related to ensuring the future health and well-being of young people [1]. Healthy lifestyles, physical activity and fitness levels in this population are in decline [2], and the COVID-19 pandemic further increased this trend [3]. The increasing prevalence of sedentary lifestyles, physical inactivity and obesity in young people highlights that they are not acquiring the necessary habits and skills to maintain a physically active lifestyle [2]. In this regard, the World Health Organisation (WHO) recommends 60 min of moderate-to-vigorous intensity physical activity (MVPA) and 3 days of muscle and bone strengthening activities per week [4] for positive outcomes on health-related variables such as physical fitness, cognitive development and mental health [5].

In the absence of government strategies to address these concerns within schools, the WHO proposed the Global Action Plan 2018–2030 for international policy systems to take educational measures to provide guidance and support for regular physical activity (PA) practice, with the goal of children growing up physically, psychologically, emotionally and socially healthy [6]. Schools provide an ideal environment to provide multidimensional health benefits to young people [7] through consistent education and exposure to different types and levels of PA [2]. School-based PA, understood as the promotion of PA during school hours [8], appears to be an effective strategy to interrupt prolonged sitting and physical inactivity, as well as to improve the MVPA and physical fitness of young people [9,10,11].

Among the strategies used to promote school-based PA, we find Classroom-based Physical Activity (CB-PA), understood as the incorporation of PA by teachers during school teaching time [7]. It is considered one of the most appropriate strategies to reduce students’ sedentary time and physical inactivity during school time, as well as to increase their cognitive involvement, academic performance and socio-emotional interaction [12,13,14]. In relation to the motor domain, different studies found that CB-PA intervention programmes had positive effects on PA levels [15,16] and physical fitness [17,18]. However, other studies found no significant changes in PA and fitness levels [19], possibly due to the characteristics of the intervention and the duration, frequency and intensity of PA applied during the programme. Therefore, further research seems necessary on the characteristics of CB-PA intervention programmes and the dose–response relationship of PA needed to achieve improvements in the motor domain of young people.

CB-PA has evolved in recent years, coming to be incorporated into multicomponent [20,21] and interdisciplinary PA programmes in combination with other teaching methodologies such as the Teaching for Personal and Social Responsibility (TPSR) model [22] to enhance multiple domains of student learning [23,24,25]. Multicomponent and interdisciplinary programmes are considered to be one of the promising approaches used to increase the PA levels [26] and physical fitness of young people [27]. These school-based interventions could help to maintain beneficial effects over time and transfer them to other contexts if they are based on theoretical models of behavioural change [26]. Taking into account that PA is influenced by multiple factors, the present educational programme is based on multiple theoretical frameworks such as the social ecological model [28], self-determination theory [29] and the theory of planned behaviour [30]. In this sense, the social ecological model, the theory of self-determination and the theory of planned behaviour establish that it is necessary to involve different agents of the school community (e.g., peers, teachers, etc.) in educational PA programmes to promote autonomous motivational processes that make it possible to achieve positive changes in health behaviour and in the intention to practice PA [26,31].

Based on the above, the main objective of the present study was to analyse the effects of a CB-PA and life skills programme on students’ PA and fitness levels. It was hypothesised that the intervention programme based on CB-PA and life skills would have positive physical consequences such as increased PA levels during school hours and improved fitness levels.

## 2. Materials and Methods

### 2.1. Design

In this study, a quasi-experimental design with repeated measures and longitudinal cut-off was carried out, as the pre-test and post-test results of a group that received a treatment under evaluation were compared with a group that did not receive any treatment in a time interval and the sample was not randomised [32]. The design of the study can be seen in the CONSORT flow diagram (Figure 1). In addition, the study presented a quantitative methodological approach, since the levels of PA and fitness of the participating groups were quantified and analysed by means of specific measures and tests at the beginning and end of the programme. The protocol of the educational intervention programme, called ACTIVE VALUES, has been detailed and published previously [22], following the Template for Intervention Description and Replication (TIDieR) [33] and the guidelines of the Transparent Reporting of Evaluation with Nonrandomised Designs (TREND) [34]. The protocol and the intervention study were approved by the Ethics Committee of the University of Murcia (3207/2021) and developed in accordance with the Declaration of Helsinki.

### 2.2. Sample

The sample in the present study was selected by convenience and accessibility, using four natural groups from the 6th grade of primary education. The sample was divided into an experimental group (EG) (*N* = 31; 15 girls; 11.87 *±* 0.56 years) and a control group (CG) (*N* = 34; 17 girls; 11.85 *±* 0.50 years), with ages between 11 and 13 years (11.86 *±* 0.53 years). The participants belonged to two public schools in the region of Murcia (Spain), with a medium-low socioeconomic level.

To minimise experimental and investigator bias, the principal investigator was not involved in the placement and collection of the devices, nor in the implementation of the CB-PA + TPSR-based programme.

#### Eligibility Criteria

The inclusion/exclusion criteria were:Students enrolled in the sixth year of elementary education.Attend school more than 80% of the time.Complete the measurements (PA and fitness) in the pre-test and post-test.Not having been diagnosed with special educational needs.No partial or chronic problems (e.g., heart disease, diabetes, asthma, injuries, etc.) that could prevent participation in data collection or programme activities.

### 2.3. Instruments

The following variables were measured to assess the well-being and physical mastery of the participants: (1) PA levels and (2) fitness.

#### 2.3.1. Levels of PA

The measurement of PA levels was performed with ActiGraph GT3X and wGTX-BT accelerometers (Actigraph, Pensacola, FL, USA), since they have triaxial measurement of acceleration [35] and have demonstrated validity and reliability for measuring this variable in the study population [36,37,38,39,40]. These devices were programmed with 60 Hz frequency (ActiGraph wGTX-BT) and a one-second epoch (ActiGraph GT3x) to collect data every day of the week during school hours (9:00 a.m. to 14:00 p.m.; school-based PA). Data were recorded in the week prior to the start of the intervention (pre-test, week 0) and in the last week of the intervention programme (week 16). Taking into account the recommendations to place the accelerometers in areas close to the centre of body mass [41,42,43], the devices were placed on the students’ right hips using an elastic belt [44]. In addition, the participants recorded, in a weekly diary, the activities they performed during recess and physical education [16].

Data were downloaded and analysed using ActiLife software (ActiGraph, version 6.8.0, Pensacola, FL, USA). Data recording was considered valid when it collected equal to or greater than four hours per day for four days per week [16]. The cut-off points established by Evenson et al. [45] in children and young people were selected to analyse the time of PA measured by the accelerometers, using the unit of measurement counts ×min^−1^: (a) sedentary (0–100), (b) light (100–2295), (c) moderate (2296–4012), and (d) vigorous (4013>). PA levels were calculated per minute/school day by dividing each PA level by the valid number of days recorded [18]. MVPA was obtained by adding up the minutes of moderate and vigorous PA [46]. To assess the impact of the programme, different moments throughout the school day and week were evaluated [16]: (a) school/curricular time; (b) recess or lunchtime; and (c) Physical Education lessons. 

#### 2.3.2. Fitness

To assess physical condition, anthropometry and basic physical capacities, the students were measured before (pre-test) and after (post-test) the intervention. The fitness assessment protocol followed the approach used in the HELENA (Healthy Lifestyle in Europe by Nutrition in Adolescence) study [47]. Three evaluators at the same time, including the Physical Education teacher, assessed these variables:Anthropometry and body composition

The Seca^®^ 876 scale and a Seca^®^ 220 telescopic height measuring instrument (Seca, Hamburg, Germany) were used to measure the weight and height of the participants. In the weight measurement, students were asked to wear light clothing and to remove footwear and other accessories that could influence the assessment. In the height measurement, students were asked to stand barefoot, upright and immobile next to a wall, with their heels together, arms extended along the body and looking straight ahead in sagittal position. Each parameter was measured twice to confirm the mean value of the measurement [48]. 

The body composition of the participants was calculated using the body mass index (BMI) formula: weight difference by height squared (kg/m^2^). The international BMI cut-off points for classifying children according to sex and age were [49]: (a) normo-weight, children with BMI values corresponding to an adult BMI below 25; (b) overweight, children with BMI values corresponding to an adult BMI between 25 and 30; and (c) obese, children with BMI values corresponding to an adult BMI above 30 [50].

Fitness level

The ALPHA-Fitness battery was used to measure students’ fitness levels [48], with the addition of a test from the EUROFIT battery [51] to assess flexibility. To ensure student safety, American College of Sports Medicine guidelines [52] and the established protocol for these test batteries [47,48,51] were followed. In addition, participants were briefed on the protocol to ensure greater success in the data collection process [19]. Data collection was conducted in the school gymnasium to maximise student safety and avoid falls due to slips [48]. The cut-off points for the physical fitness tests were obtained from the European HELENA [47] and EUROFIT [53] studies for boys and girls aged 11–13 years. The tests were evaluated following the order of the HELENA protocol [47]:Lower-limb explosive strength: the long jump test with feet together was used to measure the explosive strength of the lower body. Students were instructed to stand behind the jump line with their feet shoulder-width apart and perform a horizontal jump to reach the maximum possible distance, which was recorded in centimetres. The students performed two jumps, with 30 s of recovery between jumps to minimise the effect of fatigue [19]. The jump with the greatest distance was recorded. A Model 74-Y100M tape measure (CST/Berger, Chicago, IL, USA) was used to measure the distance of the jump.Speed/agility: the 4 × 10 meters speed test was used to evaluate the coordination, agility and speed of the participants. The students had to run and turn at maximum speed for four repetitions of 10 m distance. Each participant had two attempts with a 60-s rest between attempts. The best result obtained was recorded. The evaluators measured the test in seconds with a hand-held stopwatch (HS-80TW-1EF, Casio, Tokyo, Japan).Flexibility: the sit-and-reach test of the EUROFIT battery [51] was used to assess students’ flexibility [54,55]. Participants had to sit barefoot in front of a box (Baseline Sit n’ Reach Trunk Flexibility Box, Fabrication Enterprises Inc, Elmsford, New York, NY, USA) with their legs extended and with the soles of their feet in full contact with the wall of the box. After that, they flexed their trunk forward without bending their legs and extending their arms to try to carry the ruler as far as possible. The highest position that the students reached and were able to maintain for at least two seconds was recorded. The greatest distance reached, in centimetres and millimetres, of the two attempts made by each participant was recorded.Aerobic capacity: the 20 m out-and-back running test was used to assess aerobic capacity [56]. Participants had to run between two lines located 20 m apart and make directional changes at the pace set by audio signals that were emitted by a portable audio system (Behringer EPA40, Burgebrach, Germany) and USB player (Hayabusa, Toshiba, Tokyo, Japan) that had the test protocol [19]. The test started at a signal speed of 8.5 km/h, which increased by 0.5 km/h every minute. The participants were stopped when they were not able to keep up with the audio signal (fatigue) or to reach one of the lines for the second consecutive time. This test had only one attempt and the last half-stage completed was recorded.

### 2.4. Procedures

An educational programme called “ACTIVE VALUES” was designed and implemented based on the theoretical frameworks of the social ecological model, the self-determination theory and the theory of planned behaviour, as well as effective strategies to promote life skills and increase PA during school hours. Following the social ecological model and adopting an interdisciplinary approach to school, as previously proposed by other studies [26], the “ACTIVE VALUES” educational programme aimed to empower and support students to create their own responsible and autonomous habits in order to be physically active and increase their PA levels.

Based on the self-determination theory and the theory of planned behaviour, teachers who applied the interdisciplinary educational programme followed a specific training process composed of two stages [57]: (a) Basic Training, a stage in which an initial 15 h theoretical–practical course was carried out to provide the participating teachers with sufficient resources and strategies to further support needs, promote life skills and increase students’ PA levels; and (b) Continuous Professional Development, a stage in which the principal researcher followed up the intervention (once a month) through specific strategies such as training seminars, feedback and resolution of doubts [58].

The continuous professional development was complemented with a process of implementation fidelity [59] based on the observational analysis of the sessions implemented by the teachers during the intervention programme [60]. For this purpose, two sessions were recorded every 2–3 weeks and analysed by the research group using the tool for evaluating responsibility-based education and PA in the classroom [23,24]. Following the observational analysis, feedback reports were written by the principal investigator and shared with the teachers to provide support and guidance during the intervention [58]. In addition, teachers were invited to evaluate their own performance in implementing the educational programme after each school day to reflect on the strengths and weaknesses they were encountering in the intervention [61]. All these aspects were pooled in the training meetings/seminars held by the principal investigator with the EG teachers (once a month) to ensure greater adherence and fidelity to the intervention programme.

The intervention programme was applied for four months (60–90 min sessions; 2–3 times per day) in different subject areas, following the curriculum of the educational centres and the content of the Spanish educational legislation [62]. The interdisciplinary educational programme consisted of the application of different strategies to foster CB-PA, support needs and life skills in students, following proposals made in previous studies [22,24,25,26]. The teachers incorporated CB-PA through the methods proposed by Watson et al. [7]: (a) PA of short duration and any level of intensity (mainly MVPA) to break with the teaching dynamics, as a time-out from homework and to reduce sedentary lifestyles and physical inactivity of students (active breaks); and (b) PA directly related to curricular content (e.g., counting in one school subject) (physically active learning). Support needs and life skills were incorporated through the flexible application of the fundamental TPSR strategies for the levels of responsibility [63]: (1) respect for the rights and feelings of others; (2) participation and effort; (3) personal autonomy; (4) helping others and leadership; and (5) transfer to life outside school. The levels were approached in a progressive and interactive manner [64] to meet support needs and develop life skills related to habits of personal and social responsibility. The transfer level was used to teach and give enough tools to students to put what they had learned outside the school context (e.g., taking an active rest when sitting for a long time doing homework, doing PA autonomously, helping and encouraging family members to increase movement at home) into practice. 

The intervention programme sessions followed the TPSR session structure [63], adapted to the multidisciplinary context [22]: (1) Awareness-raising—teachers welcomed students and presented the goals to be achieved in the session related to PA and life skills (setting expectations); (2) Active responsibility—teachers set comprehensive tasks to promote CB-PA, support needs and life skills (examples of tasks can be seen in Table 1), as other studies have previously done [24,25]; (3) Group meeting—teachers proposed a discussion through an open question for students to reflect and share feelings, perceptions and opinions about their learning; and (4) Self and co-evaluation—teachers allowed students to evaluate their own, their classmates and the teacher’s performance using the thumb technique.

### 2.5. Statistics

The study variables were characterised through descriptive statistics of frequency and percentage. The Lilliefors and Shapiro–Wilk statistical tests were used to test the normality of the data, since the study sample had less than 50 participants in each group [65]. The analysis showed a non-normal distribution of the data (*p* < 0.05); therefore, the participant groups were compared (intergroup analysis; control vs. experimental) using the Mann–Whitney U statistical test comparing the variables between the control and experimental groups. Subsequently, the Wilcoxon statistical test was used to evaluate the evolution of each group (intragroup analysis) before and after the intervention (pre-test and post-test). Finally, the effect size was calculated to check the magnitude of the intergroup and intragroup differences, following the effect size values proposed by Cohen [66]: small (*d* = 0.20), medium (*d* = 0.50) and large (*d* = 0.80). Statistical analyses were performed with the statistical packages IBM SPSS 25.0 and G*Power 3.1.9.7.

## 3. Results

### 3.1. PA Levels

Table 2 shows the means and standard deviations of the PA levels of the groups participating in the study (CG and EG) as a function of the time of the school day and the intervention time (pre-post). The *p*-values obtained with the Mann–Whitney U-test revealed no significant differences (*p* < 0.05) between groups in the pre-test. Therefore, the CG and EG had a similar level before the start of the intervention programme.

Regarding pre–post intragroup differences, the Wilcoxon rank test revealed that CG significantly increased sedentary time (*p* = 0.016) during Physical Education classes and significantly reduced moderate PA (*p* = 0.004), vigorous (*p* = 0.041) and MVPA (*p* = 0.012) in Physical Education time. The EG significantly reduced the total sedentary time during school hours (*p* = 0.000), recess (*p* = 0.018) and Physical Education classes (*p* = 0.022), while it significantly increased the levels of (a) light PA during school hours (*p* = 0.036), at recess (*p* = 0.025) and Physical Education (*p* = 0.030); (b) moderate PA at school time *(p* = 0.000); (c) vigorous PA at school time (*p* = 0.000) and Physical Education (*p* = 0.012); and (d) MVPA at school time (*p* = 0.000), at recess (*p* = 0.035) and Physical Education (*p* = 0.033).

In the last week of intervention (post-test), the Mann–Whitney U-test revealed statistically significant intergroup differences in favour of the EG in light (*p* = 0.036), moderate (*p* = 0.000), vigorous (*p* = 0.000) and MVPA (*p* = 0.000) levels during weekly school hours. In addition, significant differences were found in favour of this group for the variable of light PA during weekly recess time (*p* = 0.042). The variable weekly sedentary time during school hours (*p* = 0.000) showed significant differences in favour of the CG.

### 3.2. Fitness Level

Table 3 shows the means and standard deviations of the CG and EG fitness levels at baseline and at the end of the intervention. As with the PA levels, the Mann–Whitney U-test revealed no significant differences (*p* < 0.05) in the pre-test, so there was homogeneity between groups in the fitness levels at the start of the intervention programme.

The Wilcoxon test showed statistically significant intra-group differences in the EG, as they improved their levels of explosive strength (*p* = 0.000), speed/agility (*p* = 0.000) and aerobic capacity (*p* = 0.000). In the post-test, the Mann–Whitney U-test showed statistically significant intergroup differences in favour of the EG in the fitness levels of explosive strength (*p* = 0.028) and aerobic capacity (*p* = 0.038).

## 4. Discussion

Regarding the first part of the proposed objective, to analyse the effects of combining the CB-PA and life skills on students’ PA levels, the results of the study reflected that the EG obtained improvements in all levels of PA throughout the school day, decreasing the sedentary time and increasing the levels of light, moderate and vigorous PA and MVPA, while in the CG there was no change in this regard. Previous studies [15,16,67,68,69] which focused on analysing the PA levels of primary school students during school hours using accelerometers reported similar results. Specifically, Muñoz-Parreño et al. [16] reported increases in the MVPA in total school time. Different works that included CB-PA reported changes in the MVPA level throughout the school day where the active breaks were taught [69]. Other investigations, such as those by Donnelly et al. [68], Goh et al. [15] or Van de Berg et al. [70], reported a higher number of MVPA during school hours, although without specifying at what times of the school day.

On the other hand, there are studies such as the ones by Watson et al. [71] or Martin and Murtagh [72] which did not find differences between groups in MVPA levels during school hours, stating that this could be due to the fact that children compensated for PA by being less active the rest of the school day [73], and in the case of the study by Martin and Murtagh [72], the fact that the sample size was small or there was great variability. In the current study, it is interesting to highlight that although the changes occurred within the group which received the intervention, these changes were not confirmed when comparing the CG with the EG, so some of the arguments indicated by the authors previously reviewed, such as sample size, should be taken into account.

As for the recess time, only one study has been found related to this variable [16], reporting increases in the MVPA, agreeing with the results obtained in this study. In addition, higher levels of light PA and a decrease in sedentary activity were reported in the intervention group over time.

Focusing attention on the possible changes obtained in PE or other curricular subjects in which CB-PA has been included, the study by Muñoz et al. [16] did not show any kind of change in the different variables related to PA (sedentary, light, moderate, vigorous or MVPA. These outcomes were in line with those obtained in the present study, where no significant differences were found between PA levels recorded by the intervention group compared to the control group. On the other hand, in the study by Riley et al. [69], where AB were used during mathematics lesson and PA levels were measured, changes in MVPA and sedentary time were reported. The same thing was presented by Norris et al. [67], who report changes in PA levels during classes (in which the content of the curriculum was taught). Therefore, the use of CB-PA could have a positive consequence in different curricular subjects but not in PE, which by its own character is a subject that contains PA.

Regarding the second part of the proposed objective, to analyse the effects of combining the CB-PA and life skills on students’ fitness levels, no changes in the BMI of the students were obtained, as was the case with Drummy et al. [74], who applied three active breaks of 5 min a day for 12 weeks in primary school students. On the other hand, other works that have also considered this variable have reported differences when compared to the control group, with small increases in BMI, as is the case for Donnelly et al. [68], with active breaks in the classroom and 90 min of PA per week, and for Li et al. [75] with 100 min of PA per week. 

In the present research, improvements in explosive strength, speed/agility and aerobic capacity were obtained at the level of the EG over time, as well as when comparing the results with the CG (except for speed/agility). Contrasting these results with those of other studies focused on students in the primary school stage, we found the work by Mendoza-Muñoz et al. [17], which after applying a 4-week programme of active breaks reported improvements in the cardiorespiratory capacity and speed/agility of their students. Only in one other study, although in this case focused on the secondary school stage where active classes with bike tables were used, were improvements obtained in physical condition at the cardiovascular level [76].

The rest of the studies that have assessed physical condition have not reported improvements at the level of aerobic capacity, as is the case for Van de Berg [70], who applied a programme of active breaks of 10 min a day for 9 weeks, or in González-Fernández et al. [19], studying secondary school students with two breaks a day of approximately 10 min each for 8 weeks. As previous studies have shown [7,9,77], the amount, frequency, duration and intensity (dose–response relationship) of PA in intervention programmes based on CB-PA are factors that can directly influence the achievement of positive and significant results in health-related variables such as PA level and physical fitness.

Two strengths of this study should be highlighted; firstly, the consideration of both the variables of PA levels and fitness at the same time. This is because intensity influences the results obtained and is essential for improvements in physical condition, as suggested by González-Fernández et al. [19]. There are several studies mentioned in this paper that have analysed one or the other, but not both at the same time. Indeed, very few studies found have evaluated both variables in the primary stage [68,70,74]. Secondly, it should be noted that in this study an interdisciplinary educational programme was used, whereby the CB-PA were combined with life skills (TPSR) to create responsible and autonomous habits of PA practice in students, and transfer them to different contexts of life (social and family). Taking into consideration the social ecological model, self-determination and planned behaviour theories, this interdisciplinary educational programme could be more effective, contributing to students acquiring the knowledge and skills necessary to create and maintain healthy lifestyle habits through the self-regulation of their own practice of PA.

Despite the study’s strengths, it also has several limitations. One of them is the type of research design (quasi-experimental), since the sample was not randomised and, therefore, the representativeness of the school population cannot be ensured. Another limitation is the small number of participants, which could have influenced the results in addition to not providing sufficient statistical robustness to perform multivariate tests [72,78]. An additional limitation is the measurement of PA levels during school hours, as the students did not carry the accelerometers outside of the school, which made it impossible to control extracurricular activities and verify the effects of the programme on the active behaviour of the students in their daily lives.

For these reasons, future studies should be aimed at designing randomised clinical trials with a larger number of participants in which educational programmes based on the combination of CB-PA and life skills are implemented to improve the quality of life of young people and make new generations aware of the importance of creating and maintaining responsible and autonomous habits of health and PA. In addition, prospective research should consider longitudinal interventions in which PA levels are measured outside of school hours to test the effects of the intervention programme on achieving WHO recommendations [4] and transferring healthy habits to different contexts of life (social and family). Lastly, future research could evaluate other variables related to life skills (e.g., problem solving, interpersonal relationships, etc.) and health (e.g., body folds, adiposity, heart rate, etc.), as well as the maintenance of these variables after completing the intervention by performing a test–retest.

## 5. Conclusions

The study results suggest that applying a programme based on CB-PA and life skills for 16 weeks had a positive effect on the levels of PA during school hours and the fitness of the students. These findings show the importance of implementing interdisciplinary educational programmes based on active methodologies to achieve the multilateral development of young people and improve their quality of life and motor performance. In addition, this study stands out for providing the reference PA levels to carry out a monitoring plan and achieve improvements in the physical condition of the students, thus contributing to locate the dose–response relationship of CB-PA in future studies. Some practical applications of the findings could include pedagogical tasks related to life skills during the curricular lessons that might create more adherence to PA, such as students creating a dance to learn body parts in a foreign language, as at the same time they are encouraging participation and autonomy life skills.

However, it is suggested that future research proposes longitudinal experimental designs (e.g., one academic year) with a follow-up and fidelity plan for the intervention to analyse with greater precision the evolution and transfer of students’ active behaviour to different environments (school, social and family), and moments (before, during and after the intervention). In this way, studies could evaluate the levels of PA outside the school environment to deepen the holistic understanding of the effects of the programme in relation to the achievement of the recommendations of daily PA proposed by the WHO and the third of the Sustainable Development Goals adopted by all United Nations Member States. In addition, it is suggested that future studies analyse the characteristics of the intervention such as the type (e.g., active breaks, physically active learning, gestures, etc.), frequency, duration and intensity of the CB-PA, as well as the strategies for the promotion of life skills (e.g., levels of responsibility, autonomy, transfer, empowerment, etc.) to better understand the effects of the programme on different learning domains.

## Figures and Tables

**Figure 1 healthcare-11-01256-f001:**
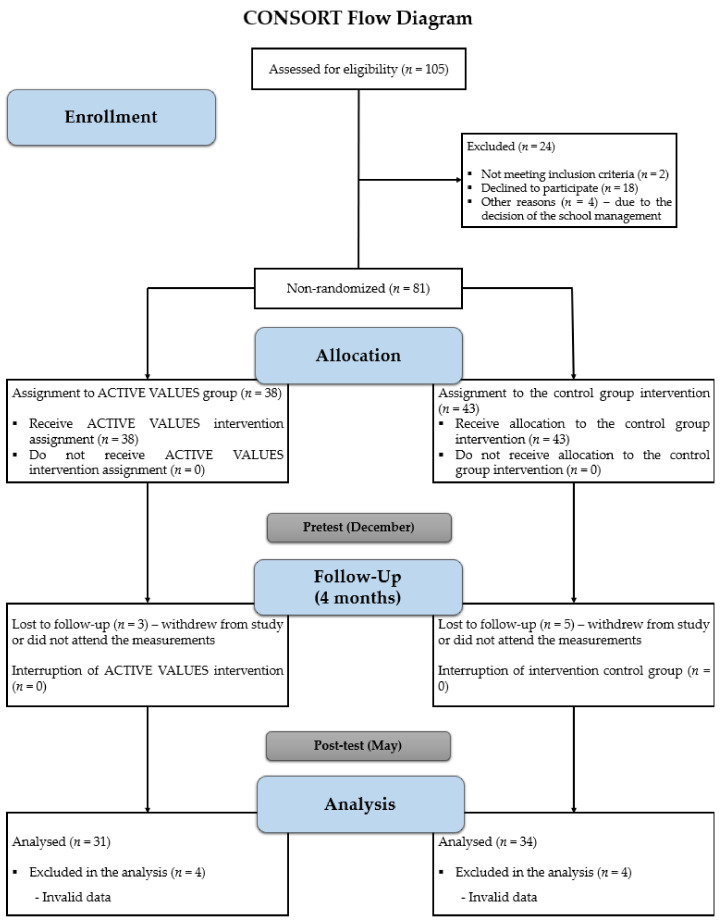
CONSORT flow diagram.

**Table 1 healthcare-11-01256-t001:** Pedagogical tasks carried out during the intervention programme.

Classroom-Based Physical Activity	Example of Tasks
Active Breaks	High intensity physical activity routine with 20” of movement (e.g., getting in and out of the chair + multi jumps + skipping) and 10” of rest (e.g., military march) (×3)
	Dance activities through the use of interactive videos (e.g., just dance)
	Leaving the classroom, running around the schoolyard and returning to class
	Exercises for joint mobility, coordination and stretching
Physically Active Learning	Solving mathematical problems by physical motion (mathematics)
	Represent and classify animals with movement according to their diet (Natural Sciences)
	Socio-dramatic representation of life skills related to levels 1 and 4 of the TPSR (Spanish Language)
	Students create a dance (TPSR levels 2 and 3) to learn body parts in the English language

Note: TPSR = Teaching Personal and Social Responsibility.

**Table 2 healthcare-11-01256-t002:** Differences in PA levels between before and the last week of the intervention programme according to group.

		Pre-Test (Week before the Start of Intervention—Week 0)				Post-Test (Last Week of Intervention—Week 16)				Pre–Post Intra-Group Test Differences			
		Control	Experimental			Control	Experimental			Control		Experimental	
School Activity (Time)	PA Levels	Mean ± SD	Mean ± SD	*p*	*d*	Mean ± SD	Mean ± SD	*p*	*d*	*p*	*d*	*p*	*d*
School day	Sedentary (min)	237.0 ± 12.97	237.5 ± 15.21	0.773	0.04	237.6 ± 14.27	220.0 ± 18.23	0.000 **	1.07	0.305	0.04	0.000 **	1.07
	Light (min)	51.2 ± 8.54	50.8 ± 11.02	0.942	0.03	50.9 ± 9.03	56.4 ± 11.15	0.037 *	0.55	0.688	0.03	0.036 *	0.51
	Moderate (min)	7.0 ± 2.84	7.1 ± 2.88	0.906	0.03	6.9 ± 3.13	15.2 ± 5.41	0.000 **	1.88	0.416	0.04	0.000 **	1.86
	Vigorous (min)	4.8 ± 2.14	4.6 ± 2.28	0.550	0.10	4.7 ± 2.60	9.2 ± 3.20	0.000 **	1.54	0.166	0.05	0.000 **	1.66
	MVPA (min)	11.9 ± 4.93	11.7 ± 5.01	0.803	0.04	11.6 ± 5.67	24.0 ± 8.42	0.000 **	1.73	0.252	0.06	0.000 **	1.78
Recess	Sedentary (min)	17.5 ± 4.24	17.6 ± 4.41	0.813	0.01	17.6 ± 4.34	16.1 ± 4.31	0.133	0.37	0.297	0.04	0.018 *	0.36
	Light (min)	7.2 ± 2.15	7.3 ± 2.20	0.916	0.05	7.1 ± 2.16	8.1 ± 2.19	0.042 *	0.48	0.297	0.04	0.025 *	0.39
	Moderate (min)	3.1 ± 1.16	2.9 ± 1.27	0.423	0.15	3.1 ± 1.25	3.3 ± 1.12	0.382	0.19	0.467	0.05	0.084	0.28
	Vigorous (min)	2.2 ± 1.10	2.1 ± 1.20	0.559	0.06	2.1 ± 1.05	2.4 ± 1.15	0.358	0.26	0.748	0.05	0.073	0.26
	MVPA (min)	5.3 ± 2.23	5.1 ± 2.45	0.454	0.11	5.2 ± 2.28	5.8 ± 2.34	0.279	0.25	0.452	0.05	0.035 *	0.30
Physical Education	Sedentary (min)	31.4 ± 4.53	31.9 ± 5.32	0.773	0.10	31.8 ± 4.83	30.5 ± 4.47	0.303	0.27	0.016 *	0.10	0.022 *	0.27
	Light (min)	21.5 ± 2.41	21.1 ± 3.30	0.793	0.12	21.3 ± 2.42	22.2 ± 2.34	0.134	0.37	0.071	0.06	0.030 *	0.37
	Moderate (min)	4.1 ± 1.33	4.1 ± 1.08	0.618	0.01	3.9 ± 1.47	4.3 ± 1.33	0.205	0.28	0.004 **	0.11	0.150	0.21
	Vigorous (min)	3.1 ± 1.19	2.6 ± 1.13	0.145	0.39	2.9 ± 1.34	2.9 ± 1.12	0.758	0.01	0.041 *	0.09	0.012 *	0.29
	MVPA (min)	7.2 ± 2.50	6.7 ± 2.13	0.586	0.20	6.9 ± 2.80	7.3 ± 2.38	0.393	0.15	0.012 *	0.10	0.033 *	0.26

Note: * *p* < 0.05; ** *p* < 0.01; SD = Standard deviation; *d* = effect size (Cohen); PA = Physical Activity; MVPA = Moderate-to-Vigorous Physical Activity.

**Table 3 healthcare-11-01256-t003:** Differences in fitness level before and after the intervention programme according to group.

		Pre-Test				Post-Test				Pre–Post Intra-Group Test Differences			
		Control	Experimental			Control	Experimental			Control		Experimental	
Variable	Factors/ Dimensions	Mean ± SD	Mean ± SD	*p*	*d*	Mean ± SD	Mean ± SD	*p*	*d*	*p*	*d*	*p*	*d*
Fitness	BMI (kg/m^2^)	21.95 ± 1.42	22.00 ± 2.07	0.793	0.03	22.02 ± 1.44	21.93 ± 1.93	0.655	0.05	0.054	0.05	0.125	0.03
	Explosive strength (cm)	118.47 ± 11.62	116.87 ± 13.49	0.650	0.13	119.03 ± 11.79	125.94 ± 12.90	0.028 *	0.56	0.348	0.05	0.000 **	0.69
	Speed/agility (s)	14.21 ± 0.95	14.36 ± 1.05	0.490	0.15	14.27 ± 1.03	14.12 ± 1.00	0.577	0.15	0.061	0.06	0.000 **	0.23
	Flexibility (cm)	25.29 ± 3.17	24.33 ± 3.40	0.273	0.29	25.17 ± 3.24	24.45 ± 3.22	0.382	0.22	0.109	0.04	0.134	0.04
	Aerobic capacity/endurance (stages)	2.46 ± 1.36	2.37 ± 1.30	0.771	0.07	2.38 ± 1.40	3.19 ± 1.50	0.038 *	0.56	0.096	0.10	0.000 **	0.58

Note: * *p* < 0.05; ** *p* < 0.01; SD = Standard deviation; *d* = effect size (Cohen); BMI = Body Mass Index.

## Data Availability

Not applicable.

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
