# Peer review of "Impact of an Interdisciplinary Educational Programme on Students’ Physical Activity and Fitness"

_healthcare, 2023, doi:10.3390/healthcare11091256_

Round 1

Reviewer 1 Report

General comments:

The main objective of this study was to analyze the effects of a classroom-based physical activity and life skills programme on students' physical activity and fitness levels.

There are some critical issues: the MVPA minutes measured only at school are very far from the daily MVPA recommended by the guidelines, therefore any reference must be made only to school-based physical activity. Please elaborate on this aspect explicitly and explain it in the Introduction and Discussion sections.

Do you have any assumptions about the leisure time daily MVPA? Therefore, I believe it is not appropriate to use min/day for MVPA because they are referred to only 6 hours per days. It would be better to consider them as an absolute value in minutes.

The findings are interesting but it appears as though most of them are not novel.

The control group did not do any activity, so it seems obvious that only the experimental group improves physical performance. At the same time, an increase of a few minutes of MVPA significantly impacts fitness levels.

The authors should therefore be clearer as to which aspects are novel. Whether novel or not, I think there is value in publishing this data that confirms previous findings.

Although described in another paper, it would be appropriate to describe the intervention for EG in more detail.

Specific comments:

Ln 100: why specify the first goal? Where students motor proficiency was analyzed in the results section?

Ln 110: I suggest using the ± symbol for all data, especially for subsequent tables (example: data were expressed as Mean±sd)

Ln 112-3: Mean±sd for are for both groups,  EG e CG

Ln 115-6: it seems that not all the students have been evaluated

Ln 127: I think it is better to explain here why it is about “quasi-experimental” design.

Ln 132: please summarize and describe the Active Values intervention.

Ln 142: why two type of accelerometers?

Ln 151:  the activities carried out during recess and  physical education, were measured with an accelerometer or with a diary?

Ln 166 :even at the beginning of the intervention?

Ln 171: “before and after the intervention” can be delete.

Ln 176: Is there a reference for this approach?

Ln 188: “Three evaluators” at the same time for each measure or at different times?

Ln 188:  PE = Physical exercise? please define it.

Ln 248: Life skills were incorporated into the intervention program. How were they evaluated pre and post intervention?

Ln 268: Does the follow-up refer to post-intervention?

Ln 276: “ GE” change with “EG”.

Ln 280: Table 2: there are many data in the table, I suggest to express as Mean±sd and, for minutes data, only one decimal place.

Ln 281 & 304: Cohen with capital first letter.

Ln 303: Table 3: like tab 2, I suggest to express data as Mean±sd. Please define the units of measurement for: Explosive strength, Speed/agility, Flexibility, Aerobic capacity (endurance).

Reviewer 2 Report

According to the attached notes

Round 2

Reviewer 1 Report

The authors revised the text as requested.

Other necessary changes

Ln 28: "UN" this acronym corresponds to Unitesd Nations?

Ln 53: delete "-based"

Ln 76-77 and 81-83: The two sentences are quite repetitive. I suggest revising the hypotheses by focusing on the two objectives ( "first of the proposed objectives" ln 331 and "second of the proposed objectives" ln 368) of the study

Ln 106: add "years" after 0,53

Ln 168: replace "cm" with "m" and the "2" in superscript

Ln 249-263: check the type and size of the ref. numbers in the brackets

Table 3: replace "centimetres" with "cm"  and "second" with "s"

Reviewer 2 Report

The proposed adjustments were made, however, it still lacks the presentation of absolute and statistical values in the abstract. In the rest, I consider the manuscript in conditions to be published.
